# Impact of High Glucose on Ocular Surface Glycocalyx Components: Implications for Diabetes-Associated Ocular Surface Damage

**DOI:** 10.3390/ijms232214289

**Published:** 2022-11-18

**Authors:** Judy Weng, Steven Trinh, Rachel Lee, Rana Metwale, Ajay Sharma

**Affiliations:** Chapman University School of Pharmacy, Chapman University, Irvine, CA 92618, USA

**Keywords:** diabetes mellitus, glycocalyx, ocular surface, cornea, conjunctiva, glycosyltransferases, hyperglycemia

## Abstract

Diabetes mellitus causes several detrimental effects on the ocular surface, including compromised barrier function and an increased risk of infections. The glycocalyx plays a vital role in barrier function. The present study was designed to test the effect of a high glucose level on components of glycocalyx. Stratified human corneal and conjunctival epithelial cells were exposed to a high glucose concentration for 24 and 72 h. Changes in Mucin (MUC) 1, 4, 16 expression were quantified using real-time PCR and ELISA. Rose bengal and jacalin staining were used to assess the spatial distribution of MUC16 and O-glycosylation. Changes in the gene expression of five glycosyltransferases and forty-two proteins involved in cell proliferation and the cell cycle were also quantified using PCR and a gene array. High glucose exposure did not affect the level or spatial distribution of membrane-tethered MUC 1, 4, and 16 either in the corneal or conjunctival epithelial cells. No change in gene expression in glycosyltransferases was observed, but a decrease in the gene expression of proteins involved in cell proliferation and the cell cycle was observed. A high-glucose-mediated decrease in gene expression of proteins involved in cellular proliferation of corneal and conjunctival epithelial cells may be one of the mechanisms underlying a diabetes-associated decrease in ocular surface’s glycocalyx.

## 1. Introduction

Poorly controlled hyperglycemia in patients with diabetes mellitus causes damage to many organs, including the eye. Retinopathy and cataracts are well-known detrimental effects of diabetes mellitus on ocular tissue, and their etiological mechanisms have been extensively investigated. Accumulating evidence over the past few years has demonstrated that diabetes mellitus also causes damage to the ocular surface, resulting in dry eye disease and diabetic keratopathy [1,2,3,4,5]. These detrimental effects of diabetes mellitus on the ocular surface are likely mediated by multiple mechanisms, including oxidative stress, inflammation, the neuropathy of the lacrimal gland and cornea, altered tear growth factor milieu, and corneal epithelial cell abnormalities [1,2,3,4,5]. 

Due to its location, the ocular surface is inherently vulnerable to the risks posed by environmental microbes, allergens, and pollutants. Both corneal and conjunctival epithelial cells confer a strong barrier function for the protection of the ocular surface [6,7,8]. Epithelial defects, recurrent erosions, corneal ulcers, corneal edema, superficial punctate keratitis, and impaired corneal epithelial wound healing in patients with diabetic keratopathy render the ocular surface susceptible to infections [1,2,3,4,5]. However, it is worthwhile to note that diabetes mellitus patients without these overt features of epithelial cell loss also show compromised epithelial barrier function and an increased incidence of corneal and conjunctival infections [9,10,11,12,13,14]. 

Furthermore, data from our lab and from other groups has also demonstrated that an in vitro exposure to a high glucose concentration can cause the impairment of the ocular surface’s epithelial cell barrier function [15,16]. These observations raise the possibility that diabetes-associated hyperglycemia can potentially cause the impairment of the ocular surface’s barrier function, even in the absence of overt epithelial defects. However, the mechanism by which a high glucose level can potentially compromise the ocular surface’s barrier function in the absence of epithelial defects remains largely unexplored. 

The ocular surface’s epithelial cell barrier primarily comprises of two major components: the glycocalyx and intracellular tight junctions [8,9,10,17]. While tight junctions prevent the paracellular passage of noxious agents, the ocular surface glycocalyx has been shown to confer a robust transcellular barrier function [8,9,10,17]. A previous study from our lab has already tested the effect of a high glucose level on corneal and conjunctival epithelial cell tight junction proteins [15]. The ocular surface glycocalyx is a glycan-rich network comprising heavily O- and N-glycated mucins [18,19,20]. Stratified corneal and conjunctival epithelial cells express the membrane-tethered mucins MUC 1, 4, and 16 on their apical surface. O-glycans are attached to the hydroxyl group of the serine and threonine residues present on these membrane-tethered mucins, while N-glycans form an amide linkage with the asparagine residues [18,19,20]. These glycan side chains primarily comprise galactose, N-acetyl galactosamine, and N-acetylglucosamine with a terminal sialic acid. The biosynthesis and assembly of these O-glycans in corneal and conjunctival epithelial cells are primarily mediated by five glycosyltransferases: polypeptide GalNAc-transferase (ppGalNAc-T), core 1 β1,3-Gal-transferase (C1GalT), core 2 β1,6-GlcNAc-transferase (C2GnT), β1,3-Gal-transferase (β3Gal-T), and β1,4-Gal-transferase (β4Gal-T). The sialyation of terminal galactose residues imparts a negative charge to the glycocalyx. These glycosylated mucins are cross-linked by multimeric galectin-3 to form a continuous network with a net negative charge, which provides a very effective barrier function [21]. Finally, ocular surface epithelial cells are constantly renewed, and their normal proliferation is also critical to maintaining a healthy glycocalyx.

Data from our laboratory has shown that both type 1 and type 2 diabetes mellitus cause a significant decrease in the corneal glycocalyx. The degree of reduction is more severe in type 1 diabetes mellitus, which is associated with a higher level of hyperglycemia [22]. High glucose levels can affect the gene expression of a variety of genes through transcriptional and epigenetic mechanisms [23,24,25]. These observations raise the possibility that high glucose exposure may cause changes in the gene expression of membrane-tethered mucins, glycosyltransferases, and proteins involved in the epithelial cell cycle, thus likely impacting glycocalyx integrity. Furthermore, glucose-mediated osmotic stress can potentially affect the spatial distribution of glycocalyx components. Therefore, the present study was designed to test the hypothesis that high glucose exposure-mediated alteration in ocular surface glycocalyx components’ expression or spatial distribution may be a possible mechanism underlying diabetes-associated compromised glycocalyx and barrier functions. 

## 2. Results

### 2.1. Effect of a High Glucose Level on Membrane-Tethered Mucins’ Gene and Protein Expression in Stratified Human Corneal and Conjunctival Epithelial Cells 

To test the effect of high glucose exposure on ocular surface mucins’ gene and protein expression, stratified human corneal and conjunctival epithelial cells were exposed to two glucose concentrations (15 mM and 30 mM) for 24 and 72 h and the gene and protein expressions were quantified using real-time PCR and ELISA, respectively. For the corneal epithelial cells, a 15 mM glucose exposure for 72 h caused a notable increase in MUC 1, MUC 4, and MUC 16 gene expression, although the increase was not statistically significant. For the conjunctival epithelial cells, significant (* *p* < 0.05 compared to the control cells exposed to normal 5 mM glucose) increases in the gene expression of MUC 4 (24 h after 30 mM glucose exposure) and MUC 16 (24 h after 15 mM and 30 mM glucose exposure) were observed (Figure 1A,B). 

Next, we tested the effect of a high glucose level on mucin protein expression. A high glucose exposure at a 15 mM concentration caused a notable decrease in MUC 1, MUC 4, and MUC 16 protein levels in the corneal epithelial cells, but the results were not statistically significant. It is worth noting that our ELISA method did not detect MUC 1 and MUC 4 protein levels in conjunctival epithelial cell lysates. It should also be noted that a 100-fold higher level of MUC 16 was detected in conjunctival epithelial cells compared to corneal epithelial cells. Furthermore, a high degree of glucose exposure did not cause any notable change in MUC 16 protein expression in conjunctival epithelial cells (Figure 2A,B).

### 2.2. Effect of a High Glucose Level on MUC 16 Barrier Function of Stratified Human Corneal and Conjunctival Epithelial Cell Using Rose Bengal Exclusion Assay

Rose bengal is an organic anionic dye that is widely used to assess epithelial damage of the ocular surface [26,27]. Previous studies have shown that ocular surface epithelial cells, upon their expression of MUC 16 and the O-glycosylated Galβ1-3GalNAc oligosaccharides side chain (T antigen), are resistant to rose bengal staining [26,27]. Thus, to assess the effect of high glucose exposure on the ocular surface’s epithelial barrier function, specifically pertaining to the distribution of MUC 16 and its O-glycosylated side chain, a rose bengal exclusion assay was performed on both stratified corneal and conjunctival epithelial cells. As can be seen in Figure 3B left panel, the conjunctival epithelial cells show a remarkably lower level of being stained by rose bengal compared to the corneal epithelial cells (Figure 3A left panel). This is anticipated, because the higher expression of MUC 16 in conjunctival epithelial cells as noted by our ELISA quantification will resist rose bengal penetration and staining, thus resulting in less stained area in the conjunctival epithelial cells compared to that of corneal epithelial cells. Furthermore, the representative images also show that high glucose exposure did not significantly affect rose bengal staining in either the corneal or conjunctival epithelial cells (Figure 3A,B). Consistent with the representative images, the quantification of the rose bengal-stained area using Image J, as depicted in the graphs, revealed no significant difference between the normal glucose- and high glucose-exposed epithelial cells (Figure 3A,B). 

### 2.3. Effect of a High Glucose Level on O-glycosyl Side Chains in Stratified Human Corneal and Conjunctival Epithelial Cells Using Jacalin Staining

Jacalin is a plant-based lectin that recognizes and stains the Galβ1-3GalNAc O-glycosylated linked oligosaccharides (T antigen) present on the membrane-tethered mucins of corneal and conjunctival epithelial cells [28]. Therefore, we used jacalin staining to test the effect of high glucose exposure on the spatial distribution of O-glycosylated linked oligosaccharides of the membrane-tethered mucins present on ocular surface epithelial cells. As is evident from the immunostaining images, the corneal epithelial cells (Figure 4A left panel) showed prominent evidence of being stained by jacalin, which was primarily localized around the cell perimeter, while minimal staining was noted in the central area of the cell. On the other hand, the conjunctival epithelial cells (Figure 4B left panel) showed a jacalin-stained region that was evenly distributed throughout the cell surface, and the intensity of the stain was remarkably higher compared to the staining of corneal epithelial cells. In addition, the immunostaining images (Figure 4A,B) show that the high glucose exposure did not significantly affect the O-glycosylated linked oligosaccharides present on the membrane-tethered mucins in either the corneal or conjunctival epithelial cells, as is evident by the jacalin-stained area. Furthermore, the graph depicting the quantification of the jacalin-stained area of conjunctival epithelial cells using Image J (Figure 4B) revealed no significant difference between the normal glucose and high glucose-exposed conjunctival epithelial cells. For the corneal epithelial cells, the images were subjectively graded by a blinded observer, and no major difference in the jacalin-staining pattern or distribution was noticed between the corneal epithelial cells exposed to a normal glucose level compared to the corneal epithelial cells exposed to a high glucose level. 

### 2.4. Effect of a High Glucose Level on Glycosyltransferases’ Gene Expression in Stratified Human Corneal and Conjunctival Epithelial Cells 

Glycosyltransferases are a family of enzymes responsible for mediating the O-glycosylation of the side chains attached to membrane-tethered mucins [19,20]. There are four possible core O-glycosylation side chains (core 1–4) containing distinct combinations of oligosaccharides present on mucins. Only core 1 and core 2 O-glycosylation side chains are expressed on the membrane-tethered mucins of corneal and conjunctival epithelial cells [19,20]. The oligomerization of these core 1 and core 2 O-glycosylation side chains is mediated by glycosyltransferases’ core 1 synthase, glycoprotein-N-acetylgalactosamine 3-beta-galactosyltransferase 1 (C1GalT1), core 2 beta-1, 6-N-acetylglucosaminyltransferase (C2GnT), polypeptide N-acetylgalactosaminyltransferase (ppGalNAC-T), beta 1,3-galactosyltransferase (β3Gal-T), and beta 1,4 galactosyltransferase (β4Gal-T). Therefore, we tested the effect of a high glucose exposure on the gene expression of these five glycosyltransferases in stratified human corneal and conjunctival epithelial cells. Both stratified human corneal and conjunctival epithelial cells showed expression of ppGalNAC-T, C1GalT1, and β4Gal-T, with corneal epithelial cells also showing an expression of β3Gal-T (Figure 5A,B). High glucose exposure did not cause any notable change in the gene expressions of any of these glycosyltransferases for either the corneal or conjunctival epithelial cells. 

### 2.5. Effect of a High Glucose Level on Expression of Genes Involved in Cell Cycle in Stratified Human Corneal and Conjunctival Epithelial Cells 

The suppression of ocular surface epithelial cell proliferation can lead to an overall decrease in ocular surface glycocalyx, even in the absence of any change in the individual components of the glycocalyx. Therefore, we next tested whether glucose can potentially modulate the expression of genes involved in cellular proliferation. To test the effect of glucose exposure on the cell proliferation of ocular surface epithelial cells, we used an array of 42 genes that are involved in the cell cycle stages of the G1 and G1/S transition phases (ANAPC2, CCND1, CCNE1, CDC34, CDK4, CDK6, SKP2, CDKN1B, CDKN3, and E2F1), S phase and DNA replication (ABL1, CDC6, MCM2, MCM3, MCM4, MCM5, and WEE1), G2 phase and the G2/M transition (BCCIP, BIRC5, CCNA2, CCNB1, CCNG1, CCNH, CCNT1, CDC25A, CDK5R1, CDK5RAP1, CDK7, CDKN3, CKS1B, CKS2, GTSE1, KPNA2, MNAT1, and SERTAD1), M phase (AURKB, CCNB2, CCNF, CDC16, CDC20, CDC25C, CDC6, CDK1, and STMN1), the cell cycle checkpoint and cell cycle arrest (CDC34, CDK1, CDKN1B, CDKN3, and WEE1), and the regulation of the cell cycle (ABL1, ANAPC2, BCCIP, CCNB1, CCNB2, CCND1, CCNE1, CCNF, CCNH, CCNT1, CDC16, CDC20, CDC25C, CDC6, CDK1, CDK4, CDK5R1, CDK6, CDK7, SKP2, CDKN1B, CKS1B, E2F1, and WEE1). As is evident from Figure 6, the glucose exposure for 72 h caused a decrease in the expression of many of these genes involved in cell proliferation. This decrease in gene expression was more prominent at a 30 mM concentration and a greater number of genes were affected in the conjunctival epithelial cells compared to the corneal epithelial cells.

## 3. Discussion

The prevalence of diabetes mellitus is increasing globally, with a staggering 537 million adults worldwide currently suffering from this disease [29]. The disease affects one out of seven adults in the United States, resulting in a significant morbidity due to its complications and accounting for a major financial burden on the health system [29]. Diabetes mellitus affects all parts of the eye. Diabetes-associated retinopathy and cataracts are major causes of blindness. Recent clinical reports and animal studies have shown that diabetes also significantly affects the ocular surface, causing an increased incidence of dry eye, corneal keratopathy, and keratitis [1,2,3,4,5,6]. The ocular surface epithelium seems to be especially affected by diabetes mellitus. Preclinical studies in animal models of diabetes and clinical observations in patients with diabetes mellitus demonstrate the presence of epithelial defects, impaired corneal epithelial wound healing, recurrent epithelial erosions, basement membrane thickening, and the loss of corneal and conjunctival epithelial barrier function [1,2,3,4,5,6]. 

The apical surface of the corneal and conjunctival epithelium is lined with a glycocalyx, a continuous network composed of glycosylated membrane-tethered mucins and galectin [18,19,20,21]. Forming a boundary between the ocular surface epithelium and the tear film, the glycocalyx maintains ocular surface hydration, reduces friction during blinking, and acts as a strong protective barrier, safeguarding the underlying epithelial cells against damage by outside chemicals and microbes [18,19,20,21]. Several studies have shown that diabetes mellitus causes a compromise of the ocular surface’s barrier function and increases the risk of corneal and conjunctival infections even when corneal keratopathy may not be fully apparent [9,10,11,12,13,14]. Using a mouse model of diabetes, a previous study conducted by our lab demonstrated that diabetes mellitus caused a significant decrease in the area of the corneal glycocalyx, with more damage seen with higher glucose levels associated with type 1 diabetes compared to type 2 diabetes [22]. Since the glycocalyx is a major component of the ocular surface barrier, our data suggest that a compromised glycocalyx could be one of the factors contributing to the reduced epithelial barrier function noted in diabetes mellitus. 

Diabetes mellitus triggers a variety of pathological changes that can cause the noted compromise of the ocular surface glycocalyx. An elevated blood glucose level or hyperglycemia, either due to insulin deficiency and/or insulin resistance, is a cardinal feature of diabetes mellitus. Multiple recent studies have shown that glucose can modulate the gene expression of a variety of genes either through the direct regulation of transcription or through epigenetic mechanisms [23,24,25]. Glucose can also affect the level or spatial distribution of proteins through osmotic stress. Thus, the goal of this study was to test whether a high glucose level can directly affect the expression or spatial distribution of glycocalyx components as the probable mechanism underlying a diabetes mellitus-associated compromised glycocalyx, even in the absence of an apparent keratopathy. To test the direct effect of high glucose exposure on various components of the glycocalyx, we used an in vitro-stratified culture of ocular surface epithelial cells, which was free from other confounding factors, including those posed by immune cells. Membrane-tethered mucins are important components of the glycocalyx. The results of the present study demonstrate that glucose does not directly affect the protein levels of ocular surface mucins but affects their gene expression. However, rather than causing a decrease in mucin gene expression, as would be anticipated for a compromised glycocalyx in vivo in diabetic mice as well as a modest decrease in mucin protein in vitro, glucose caused an increase in the mucins’ gene expression. It is likely that the noted increase in the MUCs’ gene expression may be a compensatory response to circumvent a high glucose-mediated decrease in MUC protein levels. MUC 16 is the largest mucin and the second largest human protein [26,30,31]. MUC 16 has been shown to be linked to the actin cytoskeleton, thus providing an adhesive function to glycocalyx. MUC 16 extends beyond other membrane-tethered mucins and its large size and glycosylation contribute to its strong barrier function [26,30,31]. Previous studies have demonstrated that MUC 16, serving as an important barrier component, prevents rose bengal penetration and islands of rose bengal-negative areas correlate with the distribution of MUC 16 in the stratified corneal-limbal epithelial cell [26,27]. Given the important role of MUC 16 in the glycocalyx’s barrier function, we specifically tested whether high glucose exposure disrupts the spatial distribution of MUC 16 using rose bengal exclusion staining. However, our data show that corneal and conjunctival epithelial cells exposed to high glucose do not show any increase in the areas stained by rose bengal, suggesting that high glucose-mediated alterations in MUC 16 are not the likely mechanism contributing to the diabetes-associated disruption of the glycocalyx or impaired barrier function.

The membrane-tethered mucins expressed on epithelial cells are highly glycosylated, resulting from post-translational modification through both O-glycosylation and N-glycosylation. O-glycosylation is the most predominant post-translational modification in ocular surface membrane-tethered mucins [18,19,20,21]. These O-glycosylated side chains provide a rigid conformation, negative charge, and an extended area to the mucins. These features are vital to the barrier function of the glycocalyx. The O-glycosylation of mucins is mediated by a family of enzymes called glycosyltransferases. The mucins on the ocular surface epithelial cells contain core 1 and core 2 O-glycosylation side chains. The oligomerization of these core 1 and core 2 O-glycosylation side chains is mediated by glycosyltransferases core 1 synthase, glycoprotein-N-acetylgalactosamine 3-beta-galactosyltransferase 1 (C1GalT1), core 2 beta-1, 6-N-acetylglucosaminyltransferase (C2GnT), polypeptide N-acetylgalactosaminyltransferase (ppGalNAC-T), beta 1,3-galactosyltransferase (β3Gal-T), and beta 1,4 galactosyltransferase (β4Gal-T) [19,20]. A decrease in the glucose-mediated gene expression of these enzymes can potentially affect the degree of mucin O-glycosylation. However, our data demonstrate that glucose does not affect the gene expression of these glycosyltransferases. Furthermore, our results also demonstrate that a high glucose level did not affect the spatial distribution or network continuity of the O-glycan side chains, as tested using jacalin staining, since no notable changes regarding jacalin staining were observed in the corneal or conjunctival epithelial cells exposed to a high glucose concentration.

The corneal and conjunctival epithelia are constantly renewing, and their normal proliferation is important to maintain a healthy glycocalyx network. Multiple published studies have shown that an in vitro exposure of corneal epithelial cells to a high glucose concentration can significantly reduce their proliferation and viability [16,32,33]. Thus, we tested whether glucose exposure can modulate the gene expression of proteins involved in the cell proliferation of epithelial cells. Indeed, the results of this study demonstrate that high glucose exposure reduces the expression of several key mediators involved in cell cycle and proliferation. The reduction in the expression of these key protein mediators of the cell cycle and proliferation may explain the well-documented decrease in glucose-mediated epithelial cell proliferation or viability. Furthermore, the noted decrease in the glucose-mediated expression of these proteins and the consequent reduction in ocular surface epithelial cell proliferation likely mediate a decrease in ocular surface glycocalyx and compromised barrier function, as noted via the high glucose exposure. 

## 4. Materials and Methods

### 4.1. Stratified Human Corneal Epithelial and Conjunctival Epithelial Cell Cultures

Telomerase-immortalized human corneal epithelial cells (provided by Dr. James V. Jester, School of Medicine, University of California, Irvine, CA, USA) and human conjunctival epithelial cells (provided by Dr. Ilene Gipson, Harvard Medical School, Boston, MA, USA) were used for the present study. Human corneal epithelial cells were cultured in keratinocyte growth medium (PromoCell GmbH, Heidelberg, Germany) supplemented with bovine pituitary extract (0.004 mL/mL), human epidermal growth factor (0.125 ng/mL), human insulin (5 μg/mL), hydrocortisone (0.33 μg/mL), epinephrine (0.39 μg/mL), transferrin (10 μg/mL), and calcium chloride (0.15 mM). Stratification was induced by plating corneal epithelial cells on 0.4 uM polyethylene terephthalate transwell membrane inserts (Sterlitech, Auburn, WA, USA) and switching to supplemented keratinocyte growth medium with a higher concentration of calcium chloride (1.15 mM). This medium was added to both the membrane inserts and bottom wells until cells reached 100% confluence. After reaching 100% confluence, medium was added only to the bottom well, leaving the cell-containing membrane inserts exposed to air for one week, which resulted in stratification [34].

Human conjunctival epithelial cells were cultured in keratinocyte serum-free medium (Gibco, Thermo Scientific, Rockford, IL, USA) supplemented with bovine pituitary extract (25 μg/mL), epidermal growth factor (0.2 ng/mL) and calcium chloride (0.4 mM). Once cells reached 50% confluence, they were cultured in a medium containing supplemented keratinocyte serum-free medium and a 1:1 mixture of F12: DMEM without calcium (Gibco-Invitrogen Corp., Rockville, MD, USA). After cells reached 100% confluency, stratification was induced by replacing media with DMEM/F12 medium supplemented with calcium chloride (1 mM), calf serum (10%), and EGF (10 ng/mL) for around 5 to 7 days [35].

### 4.2. Glucose Treatments 

To test the effect of a high glucose level, the stratified human corneal and conjunctival epithelial cells were exposed to media containing 15 mM and 30 mM glucose for 24 and 72 h. The control cells were cultured in regular media that contained 5 mM of glucose. The experiments were conducted in triplicate to obtain cells for mucins and glycosyltransferases’ gene expression, for mucins’ ELISAs, for rose bengal and jacalin staining, and for cell proliferation array. 

### 4.3. Isolation of mRNA Preparation of cDNA 

Total mRNA from control and high-glucose exposed corneal and conjunctival epithelial cells was extracted using a commercially available kit, following manufacturer’s protocol (RNeasy Mini Kit; QIAGEN, Valencia, CA, USA). Isolated mRNA was subsequently reverse-transcribed into cDNA using a commercially available kit (SuperScript III First-Strand; Invitrogen, Carlsbad, CA, USA). 

### 4.4. Mucin and Glycosyltransferase Gene Expression Quantification 

Quantification of mucins (MUC 1, 4, and 16) and glycosyltransferase enzymes, Core 1 β3-Gal-T (C1GalT1), Core 2 β6-GlcNAc-T (C2GnT), polypeptide GalNAc-T (ppGalNAC-T), and β3Gal-T and β4Gal-T) was performed using real-time PCR (BioRad, Hercules, CA, USA). β-actin was used as the housekeeping gene. Briefly, 20 μL reaction mixtures consisting of 2 μL of cDNA, 10 μL of SYBR Master Mix, 2 μL of forward primer, 2 μL of reverse primer, and 4 μL of DEPC water were run at a universal cycle (95 °C for 10 min, 40 cycles at 95 °C for 15 s, and 55 °C for 60 s) using a real-time thermocycler (Biorad, Hercules, CA, USA). Results were first normalized to β-actin for calculation of ΔCt and then normalized to control within each treatment group to calculate fold change in gene expression using ΔΔCt method.

### 4.5. Mucin Enzyme-Linked Immunosorbent Assays

Total protein extract was prepared by incubating high-glucose-exposed stratified corneal and conjunctival epithelial cells for 15 min in radioimmunoprecipitation assay (RIPA) buffer containing Halt protease inhibitor (Thermo Fisher Scientific, Rockford, IL, USA) on a shaker, followed by removal of adherent cells using a cell scraper. The resulting supernatant was collected as total lysate and total protein concentration was determined with the BCA method using a commercially available kit (Pierce BCA Protein Assay Kit; Thermo Fisher Scientific, Rockford, IL, USA). Quantification of MUC 1, 4, and 16 proteins was performed using commercially available enzyme-linked immunosorbent assay (ELISA) kits (LifeSpan Biosciences, Seattle, WA, USA). 

### 4.6. Jacalin Staining 

The cells were fixed by incubating them in methanol at room temperature for 15 min, followed by washing them in PBS three times. The cells were blocked using 1% bovine serum albumin for 30 min at room temperature. The cells were then stained with fluorescein-conjugated jacalin (1:100 dilution) for 60 min followed by five washing procedures using PBS; then, they were mounted in DAPI-containing medium. Stained cells were imaged using a confocal microscope. Image J was used to quantify the area of staining for conjunctival cells. The jacalin-stained corneal cell images were graded on a scale of 1–5 in a blinded manner. 

### 4.7. Rose Bengal Staining 

The cells were washed three times with Ca^2+^/Mg^2+^ free PBS, and then were incubated in 0.1% rose bengal solution for 2–5 min followed by being washed three times in Ca^2+^/Mg^2+^-free PBS. Stained cells were imaged using the Keyence Brightfield microscope and the area of staining was quantified using Image J. 

### 4.8. PCR Array for Quantification of Genes Involved in Cell Cycle 

A commercially available gene array was used to quantify the expression of 42 genes involved in the cell cycle using cDNA obtained from corneal and conjunctival cells exposed to high glucose level. These genes are involved in regulation of G1 phase and G1/S transition (ANAPC2, CCND1, CCNE1, CDC34, CDK4, CDK6, SKP2, CDKN1B, CDKN3, and E2F1), S phase and DNA replication (ABL1, CDC6, MCM2, MCM3, MCM4, MCM5, and WEE1), G2 phase and G2/M transition (BCCIP, BIRC5, CCNA2, CCNB1, CCNG1, CCNH, CCNT1, CDC25A, CDK5R1, CDK5RAP1, CDK7, CDKN3, CKS1B, CKS2, GTSE1, KPNA2, MNAT1, and SERTAD1), M phase (AURKB, CCNB2, CCNF, CDC16, CDC20, CDC25C, CDC6, CDK1, and STMN1), cell cycle checkpoint and cell cycle arrest (CDC34, CDK1, CDKN1B, CDKN3, and WEE1), and regulation of the cell cycle (ABL1, ANAPC2, BCCIP, CCNB1, CCNB2, CCND1, CCNE1, CCNF, CCNH, CCNT1, CDC16, CDC20, CDC25C, CDC6, CDK1, CDK4, CDK5R1, CDK6, CDK7, SKP2, CDKN1B, CKS1B, E2F1, and WEE1). 

### 4.9. Statistical Analysis

The data are presented as mean ± standard error of the mean. Statistical analysis was performed using GraphPad Prism software (GraphPad Prism, version 9; GraphPad, San Diego, CA, USA). Two-way ANOVA followed by Bonferroni post hoc test were used to analyze the data. Statical significance was determined as *p* value < 0.05. 

## 5. Conclusions

While a high glucose level does not directly affect the level or spatial distribution of membrane-tethered mucins or their O-glycan side chains, the decreased expression of key protein mediators involved in the cell cycle and proliferation and the consequent reduction in the cellular proliferation of corneal and conjunctival epithelial cells due to a high glucose level, as noted in diabetes mellitus, may be among the mechanisms underlying the diabetes-associated decrease in ocular surface glycocalyx and compromised barrier function. Thus, it is worth exploring whether diabetes-associated inflammation, the activation of the immune system, or the activation of signaling pathways associated with advanced glycation end products may be additional mechanisms contributing to diabetes-associated detrimental effects on the ocular surface glycocalyx.

## Figures and Tables

**Figure 1 ijms-23-14289-f001:**
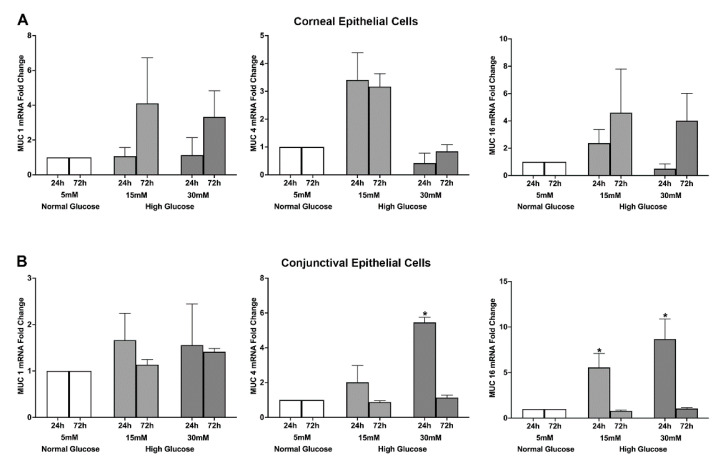
(**A**) Effect of a high glucose level (15 mM; 30 mM) on MUC 1, 4, and 16 gene expression in stratified human corneal epithelial cells. (**B**) Effect of a high glucose level (15 mM; 30 mM) on MUC 1, 4, and 16 gene expression in stratified human conjunctival epithelial cells. The data represent mean + S.E.M of cell culture experiments conducted in triplicate. * *p* < 0.05 compared to control cells exposed to normal (5 mM) glucose level.

**Figure 2 ijms-23-14289-f002:**
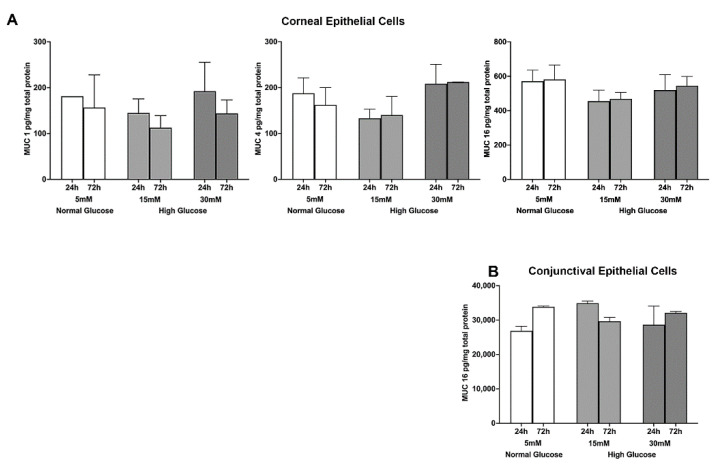
(**A**) Effect of a high glucose level (15 mM; 30 mM) on MUC 1, 4, and 16 protein expression in stratified human corneal epithelial cells. (**B**) Effect of a high glucose level (15 mM; 30 mM) on MUC 1, 4, and 16 protein expression in stratified human conjunctival epithelial cells. The data represent mean + S.E.M of cell culture experiments conducted in triplicate.

**Figure 3 ijms-23-14289-f003:**
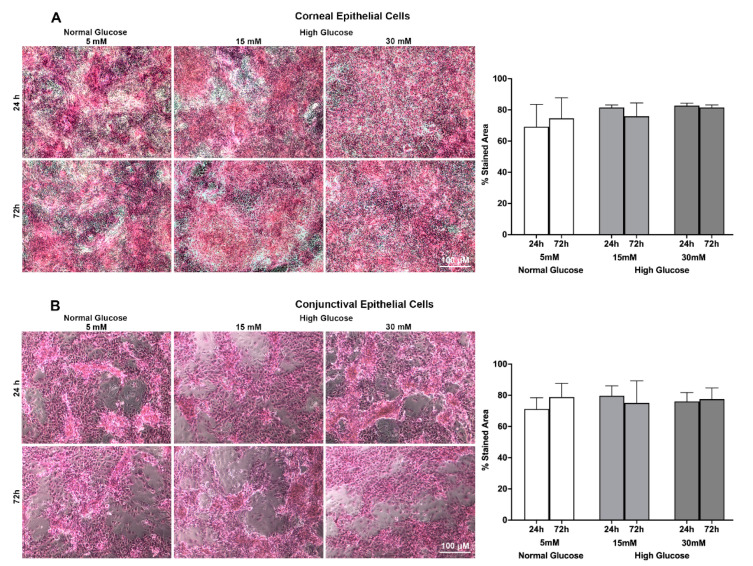
(**A**) Representative brightfield images showing effect of a high glucose level (15 mM; 30 mM) on MUC 16 barrier function as measured by rose bengal exclusion assay in stratified human corneal epithelial cells. (**B**) Representative brightfield images showing effect of a high glucose level (15 mM; 30 mM) on MUC 16 barrier function as measured by rose bengal exclusion assay in stratified human conjunctival epithelial cells. Epithelial cells with potential loss of MUC 16 are stained pink with rose bengal. Graphs show the percent-stained area as quantified using Image J software of *n* = 6 images/group. The images were obtained from cell culture experiments conducted in triplicate. Scale bar = 100 μM.

**Figure 4 ijms-23-14289-f004:**
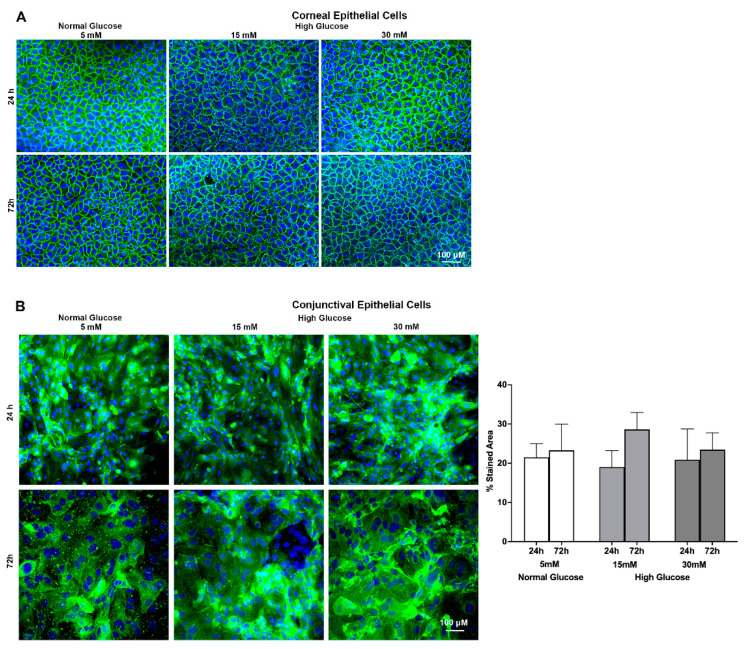
(**A**) Representative confocal microscopy images showing effect of a high glucose level (15 mM; 30 mM) on O-glycosylated Galβ1-3GalNAc oligosaccharide side chains of membrane-tethered mucins on stratified human corneal epithelial cells. (**B**) Representative confocal microscopy images showing effect of a high glucose level (15 mM; 30 mM) on O-glycosylated Galβ1-3GalNAc oligosaccharide side chains of membrane-tethered mucins on stratified human conjunctival epithelial cells. Nuclei are stained blue with DAPI, and O-glycosylated linked oligosaccharides are stained green with jacalin. Graphs show the percent-stained area as quantified using Image J software of *n* = 6 images/group. The images were obtained from cell culture experiments conducted in triplicate. Scale bar = 100 μM.

**Figure 5 ijms-23-14289-f005:**
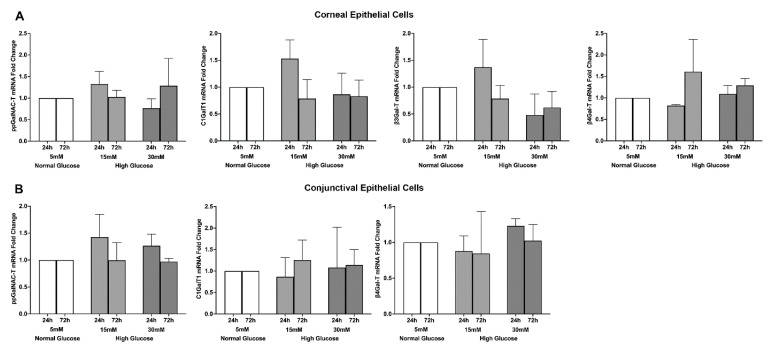
(**A**) Effect of a high glucose level (15 mM; 30 mM) on glycosyltransferase gene expression in stratified human corneal epithelial cells. (**B**) Effect of a high glucose level (15 mM; 30 mM) on glycosyltransferase gene expression in stratified human conjunctival epithelial cells. The data represent mean + S.E.M of cell culture experiments conducted in triplicate.

**Figure 6 ijms-23-14289-f006:**
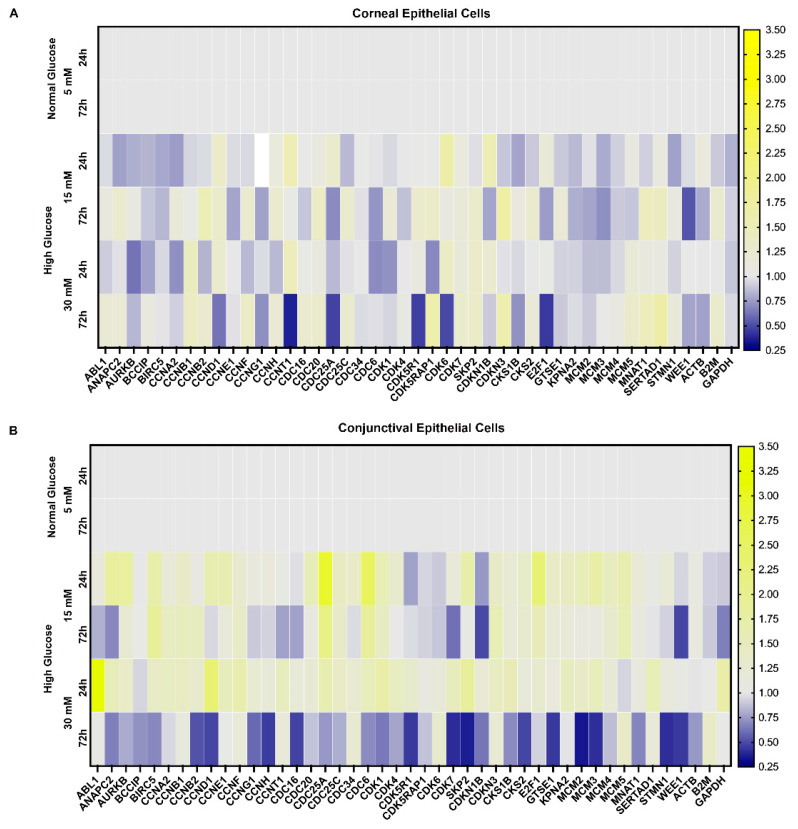
(**A**) Heat map showing changes in gene expression of proteins involved in proliferation and cell cycle in stratified human corneal epithelial cells exposed to high glucose concentration (15 mM; 30 mM). (**B**) Heat map showing changes in gene expression of proteins involved in proliferation and cell cycle in stratified human conjunctival epithelial cells exposed to high glucose concentration (15 mM; 30 mM). The proteins that show a fifty percent or higher decrease in expression are depicted in blue.

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
