# Peer review of "Impact of High Glucose on Ocular Surface Glycocalyx Components: Implications for Diabetes-Associated Ocular Surface Damage"

_ijms, 2022, doi:10.3390/ijms232214289_

Round 1

Reviewer 1 Report

This article entitled “Impact of high glucose on ocular surface glycocalyx components: Implications for diabetes-associated ocular surface damage” gives an innovative concept to recognize the ocular surface damage caused by high glucose. After reviewing the text, some points need to be addressed as the followings.

1.     In Figure 1 and 2, in conjunctival epithelial cells, the MUC genes but not proteins elevated significantly after treatment with glucose. Because the protein accumulation would follow the gene expression, it needs to be considered whether same time-duration for examining gene and protein level is reasonable or not.

2.     In Figure 6, the results from gene array revealed high glucose impaired cell cycle in corneal and conjunctival cells, however, 1, the authors did not examine whether high glucose treatment induce cell death or cell cycle impairment; 2, the cell treated with high glucose should be examined the level of cell cycle related proteins.

3.     Although the authors represented the alteration of genes in cell cycles after high glucose treated the indicated cells, it was not discussed in Discussion.

4.     As the authors mentioned in line 76-77 “… glucose-mediated osmotic stress can potentially affect the spatial distribution of glycocalyx components”, high glucose may increase the osmotic stress in cell environment. This reviewer suggested the authors use mannose as control to compare the osmotic alteration with that of glucose.

Author Response

  1. In Figure 1 and 2, in conjunctival epithelial cells, the MUC genes but not proteins elevated significantly after treatment with glucose. Because the protein accumulation would follow the gene expression, it needs to be considered whether same time-duration for examining gene and protein level is reasonable or not.

We thank the reviewer for bringing this to our attention. As per reviewer’s suggestion, this point has been added in our discussion (Page 8, Line 278-280). The significant increase in MUC 4 and 16 gene expression in conjunctival epithelial cells was noted only at 24 hours. As rightly pointed out by the reviewer, we anticipated an increase in MUC 4 and 16 protein levels at a later time point (72 hours). This is the reason that we quantified the mucin proteins at 72 hours also. However, no increase in the protein levels of MUC 4 or MUC 16 was noted at this later time point of 72 hours. Overall, it seems likely that the early time point increase in gene expression at 24 hours may partially compensate for a glucose-mediated decrease in MUC protein expression at the later time point (72 hours). We have added this consideration in discussion.

  1. In Figure 6, the results from gene array revealed high glucose impaired cell cycle in corneal and conjunctival cells, however, 1, the authors did not examine whether high glucose treatment induce cell death or cell cycle impairment; 2, the cell treated with high glucose should be examined the level of cell cycle related proteins.

We thank the reviewer for this great suggestion. Multiple studies have already shown that glucose causes a decrease in ocular surface epithelial cell proliferation and viability. Therefore, we did not repeat these experiments already published by other investigators. As per reviewer’s suggestion, we have added this information in the discussion citing these studies (reference number 16,34,35) and discussion has been updated highlighting that high glucose is known to cause a decrease in corneal epithelial cell proliferation and viability (Page 9, Line 315-317). The novel goal of the current study was to investigate whether glucose can modulate the gene expression of mediators involved in cell proliferation and cell cycle. Results of the present study demonstrate that glucose indeed modulates the expression of these mediators, which is likely responsible for the already well-documented effect of glucose on cell proliferation.

  1. Although the authors represented the alteration of genes in cell cycles after high glucose treated the indicated cells, it was not discussed in Discussion.

We thank the reviewer for this great suggestion. We have updated the discussion to include these important observations (Page 9, Line 319-326).

  1. As the authors mentioned in line 76-77 “… glucose-mediated osmotic stress can potentially affect the spatial distribution of glycocalyx components”, high glucose may increase the osmotic stress in cell environment. This reviewer suggested the authors use mannose as control to compare the osmotic alteration with that of glucose.

We presume that the reviewer is referring to mannitol, not mannose. We, in our previous publications, and also several other authors have used mannitol as a non-metabolizable osmotic control for glucose. However, in this study, glucose did not affect mucin glycosylation, as shown by the lack of high glucose exposure’s effect on rose bengal or Jacalin staining. Since glucose itself did not affect the spatial distribution of mucin glycosylation, we genuinely feel there is no need to test whether osmotic stress (using mannitol) is a probable mechanism. Had we seen this effect with glucose, it would be reasonable to investigate whether this effect of glucose is due to metabolic or osmotic effect by using mannitol as a control.

Reviewer 2 Report

The current manuscript investigates the role of high glucose on components of glycocalyx. The authors suggested that the high glucose-mediated decrease in gene expression of proteins involved in cellular proliferation of corneal and conjunctival epithelial cells, which may be one of the mechanisms underlying diabetes-associated decrease in ocular surface glycocalyx. Overall, the experiments are properly organized and their in vitro observation is clinically sound. However, some concerns listed below limit the clear narrative of the current study.

1. In the corneal epithelium, the paracellular pathway, composed of tight junctional complexes consisting of predominantly the proteins, such as occludins and claudins, is another major barrier component that contributes to the barrier function, in addition to the epithelial components. However, in the current study, there is no data shown the tight junctional complexes were involved in the high glucose -mediated regulation of barrier function in stratified human corneal and conjunctival epithelial cells. The author should provide additional data for the major tight junctional complex proteins, such as occludin and Claudin-5, to evaluate the role of high glucose exposure in the paracellular pathway.

2. In this study, the authors used the stratified human corneal and conjunctival epithelial cells for identifying the gene expression of proteins involved in cell proliferation and cell cycle in vitro experiments. However, it is not clear whether there is same gene expression pattern in the diabetic model in vivo. I suggest the authors should include animal experiments to verify their in-vitro observation. 

Author Response

  1. In the corneal epithelium, the paracellular pathway, composed of tight junctional complexes consisting of predominantly the proteins, such as occludin and claudins, is another major barrier component that contributes to the barrier function, in addition to the epithelial components. However, in the current study, there is no data shown the tight junctional complexes were involved in the high glucose -mediated regulation of barrier function in stratified human corneal and conjunctival epithelial cells. The author should provide additional data for the major tight junctional complex proteins, such as occludin and Claudin-5, to evaluate the role of high glucose exposure in the paracellular pathway.

We thank the reviewer for pointing this out. Our group has already published the manuscript that investigated the effect of high glucose exposure on various components of tight junction proteins, including the changes in both gene and protein expression of occludin, claudin 1, 2, 3, and ZO-1, 2, and 3. This manuscript is “Alfuraih S, Barbarino A, Ross C, Shamloo K, Jhanji V, Zhang M, Sharma A. Effect of High Glucose on Ocular Surface Epithelial Cell Barrier and Tight Junction Proteins. Invest Ophthalmol Vis Sci. 2020 Sep 1;61(11):3”. We have cited this study in the introduction as reference 15 and have included this information in the introduction (Page 2, Line 53-55) that our lab has previously published the effects of high glucose on corneal and conjunctival epithelial cell.

  1. In this study, the authors used the stratified human corneal and conjunctival epithelial cells for identifying the gene expression of proteins involved in cell proliferation and cell cycle in vitro experiments. However, it is not clear whether there is same gene expression pattern in the diabetic model in vivo. I suggest the authors should include animal experiments to verify their in-vitro observation.

This is again a very great suggestion. We have extensively tested the effect of diabetes mellitus in vivo on glycocalyx and also the underlying mechanisms of impaired glycocalyx using animal models of both type 1 and type 2 diabetes mellitus. We have cited this in vivo study (reference 22) in discussion multiple times. In vivo effects of diabetes mellitus are compounded by upregulation of inflammatory cytokines, involvement of immune cells and formation of AGEs, sorbitol etc. In this manuscript, we wanted to investigate whether high glucose by itself is a trigger to cause alterations in glycocalyx components without the other confounding in vivo factors.

Round 2

Reviewer 1 Report

This article has been revised according to this reviewer's comments.

Reviewer 2 Report

The authors have properly addressed my concerns in the revised manuscript. I have no further comments.